# Organic dry pea (*Pisum sativum* L.): A sustainable alternative pulse-based protein for human health

**Dil Thavarajah**[1]*, **Tristan Lawrence**[1], **Lucas Boatwright**[1], **Nathan Windsor**[1], **Nathan Johnson**[1], **Joshua Kay**[1], **Emerson Shipe**[1], **Shiv Kumar**[2], **Pushparajah Thavarajah**[1]

**1** Plant and Environmental Sciences, Pulse Quality and Nutritional Breeding, Biosystems Research Complex, Clemson University, Clemson, South Carolina, United States of America, **2** Food Legumes Research Program, International Centre for Agricultural Research in the Dry Areas (ICARDA), Amlaha, India

* dthavar@clemson.edu

**Data Availability Statement:** All relevant data are within the manuscript and its Supporting information files.

## Abstract

Dry pea (*Pisum sativum* L.) is a cool-season food legume rich in protein (20–25%). With increasing health and ecosystem awareness, organic plant-based protein demand has increased; however, the protein quality of organic dry pea has not been well studied. This study determined the genetic variation of individual amino acids (AAs), total AAs (liberated), total protein, and *in vitro* protein digestibility of commercial dry pea cultivars grown in organic on-farm fields to inform the development of protein-biofortified cultivars. Twenty-five dry pea cultivars were grown in two USDA-certified organic on-farm locations in South Carolina (SC), USA, for two years (two locations in 2019 and one in 2020). The concentrations of most individual AAs (15 of 17) and the total AA concentration significantly varied with dry pea cultivar. *In vitro* protein digestibility was not affected by the cultivar. Seed total AA and protein for dry pea ranged from 11.8 to 22.2 and 12.6 to 27.6 g/100 g, respectively, with heritability estimates of 0.19 to 0.25. *In vitro* protein digestibility and protein digestibility corrected AA score (PDCAAS) ranged from 83 to 95% and 0.18 to 0.64, respectively. Heritability estimates for individual AAs ranged from 0.08 to 0.42; principal component (PCA) analysis showed five significant AA clusters. Cultivar Fiddle had significantly higher total AA (19.6 g/100 g) and digestibility (88.5%) than all other cultivars. CDC Amarillo and Jetset were significantly higher in cystine (Cys), and CDC Inca and CDC Striker were significantly higher in methionine (Met) than other cultivars; CDC Spectrum was the best option in terms of high levels of both Cys and Met. Lysine (Lys) concentration did not vary with cultivar. A 100 g serving of organic dry pea provides a significant portion of the recommended daily allowance of six essential AAs (14–189%) and daily protein (22–48%) for an average adult weighing 72 kg. Overall, this study shows organic dry pea has excellent protein quality, significant amounts of sulfur-containing AAs and Lys, and good protein digestibility, and thus has good potential for future plant-based food production. Further genetic studies are warranted with genetically diverse panels to identify candidate genes and target parents to develop nutritionally superior cultivars for organic protein production.

**Funding:** Founding Agencies: 1. National Institute of Food and Agriculture, Organic Agriculture Research and Extension Initiative (OREI) (award no. 2018-51300-28431/proposal no. 2018-02799) and the United States Department of Agriculture, 2. National Institute of Food and Agriculture, Organic Agriculture Research and Extension Initiative (OREI) (award no. 2021-51300-34805/proposal no. 2021-02927) (DT, LB) 3. USDA National Institute of Food and Agriculture, [Hatch] project [1022664] (DT); 4. Good Food Institute (DT) 5. FoodShot Global (DT) The funders had no role in study design, data collection and analysis, decision to publish, or preparation of the manuscript.

**Competing interests:** The authors have declared that no competing interests exist.

## Introduction

The plant-based protein market has been steadily growing globally. Globally, the plant-based protein market will continue to increase to a $9.5B industry by 2025. American retail sales increased by 6% in 2021, bringing the plant-based protein market's total value to $7.4B in 2021 [1]. About 39% of Americans consume plant-based protein alternative foods due to various health and ecosystem concerns [2,3]. To meet the global demand for plant-based protein, ingredient suppliers have expanded the need for novel, pesticide-free, and gluten-free plant proteins from organic pulse crops, including dry pea (*Pisum sativum* L.). Dry peas are the most in-demand ingredient for this segment of the food industry due to their high protein (20–25%) and low fat (<1%) levels. Global dry pea production is approximately 14.6 million metric tons, and the United States of America (USA) accounted for 0.9 million metric tons in 2020 [4]. Dry pea is a vital pulse crop to provide global nutritional security, especially for protein and low digestible carbohydrates at a low cost. Canada, Russian Federation, France, China, and India have become the world's most extensive dry pea producers [4]. Certified organic dry pea production has increased globally and in US regions, but the exact statistics for the current production and harvested area for regional states are unavailable. However, US regions that have not been historically used to grow these pulse crops, including South Carolina (SC), will increase pulse production to meet the demand for plant-based protein [5,6]. Current food choices, including the "*Beyond Burger*," use dry pea as their primary protein ingredient. However, concerns related to plant-based proteins focus on amino acid (AA) balance, i.e., legume-based protein is low in sulfur-containing AAs (SAAs: cystine, Cys, and methionine, Met) and poor digestibility. The development and selection of nutritionally superior organic dry pea cultivars will bring significant economic benefits to organic growers and nutritional value to consumers.

Organic agriculture is the fastest-growing segment of US agriculture, with total sales of $9.9B in 2019 [6]. Organic grains, including corn, wheat, and soybean, accounted for $1.18B of this total, an increase of 55% from 2016 [6]. Pulse crops are an integral part of the global food system and can provide protein, low-digestible carbohydrates, and micronutrient-rich foods at a lower price than systems centered on animal proteins [5,7–9]. Certified organic dry pea production in the USA is small, with 16,666 ac in 2019 and a $5.9M value of sales from 104 USDA-certified organic farms [6]. Legume-inclusive cropping systems bring multiple benefits to organic agriculture: (1) at the food system level, pulses provide a significant amount of protein, low-digestible carbohydrates, and a range of micronutrients with low phytate for both humans and animals; (2) at the production system level, legumes fix atmospheric N and improve soil phosphorus (P), which provides economic value to organic producers, making them more suitable for low-input cropping systems and mitigating greenhouse gas emissions as a result of reduced rates of N and P fertilizer inputs; and (3) at the cropping system level, legumes can be used as diversification crops in agroecosystems (e.g., in rotation with cereals), resulting in increased cereal crop yields due to disrupted pest (disease, insect, and weed) cycles, conserved soil moisture, and improved soil health via soil microbial activity [5,10–12].

Plant-based protein represents about 60% of the total global protein consumed, with the remainder from animal sources [13]. Developing countries mainly depend on plant-based proteins, and the global population is shifting toward alternative proteins. However, the quantity of protein in plant-based foods is not the only indicator of their ability to meet the nutritional demands of growing populations; protein quality, i.e., AA balance, must be considered [14]. Animal and plant-based proteins have different AA profiles and digestibility. Humans cannot synthesize essential AAs; therefore, these must be obtained from dietary sources [15]. Cereals have low lysine (Lys) concentrations and moderate-to-high concentrations of the

SAAs, namely methionine (Met) and cysteine (Cys); however, pulses have high Lys and low-to-moderate levels of Met and Cys. Therefore, plant-based diets require cereals supplemented with Lys-rich ingredients such as dry pea [9,16]. With the increasing global population, dependence on animal sources for daily human protein requirements is not viable (e.g., higher energy and labor requirements, antibiotic resistance, and greenhouse gas emissions). Therefore, breeding traditional pulse crops for protein biofortification is essential to provide clean, allergen- and gluten-free, and highly nutritious plant-based protein to meet the world's protein demands by 2050 [9,16,17].

Biofortification is a sustainable approach to increasing nutritional quality using conventional plant breeding and genomic tools to develop staple food crops with bioavailable micronutrients [18–20]. Dry pea micronutrient enrichment has been successful over the years, and breeding programs worldwide use available tools to develop mineral- and vitamin-rich cultivars with low phytic acid [21–23]. These biofortified cultivars perform well under non-organic growing systems but have low yields and protein content when grown under an organic system without synthetic fertilizers and pesticides [5,24]. Therefore, organic nutritional breeding of pulse crops for increased protein quality is vital to overcome the issues related to growers, the food industry, and the nutrition community to meet growing consumer demand. This study aimed to evaluate if the current dry pea cultivars in production vary in protein quality (AA composition, total AA, total protein, and *in vitro* protein digestibility) in response to organic cropping systems. This study did not include the non-organic system for comparison. The objectives of this study were to assess 25 dry pea cultivars grown in two organic on-farm locations for two years to determine the genetic variation of AA profiles, total AA, protein, and *in vitro* digestibility to identify suitable cultivars for organic production with increased nutrition quality.

## Materials and methods

### Experimental details

The experimental field design was a randomized complete block design (RCDB) with 25 cultivars with two replications at two locations in 2019 and three replications at one location in 2020 (n = 175; [5]. The seeds were purchased from Pulse USA (Bismark, ND, USA), Meridian Seeds (Mapleton, ND, USA), and the Washington State Crop Improvement Association (Pullman, WA, USA). Material transfer agreements (MTAs) were signed with the seed companies before planting these cultivars in SC, USA. Detailed experimental design, agronomic details, and results (grain yield and nutritional quality) have already been published [5]. In brief, the experimental field design was a randomized complete block design (RCDB) with 25 cultivars with two replications at two locations in 2019 and three replications at one location in 2020 (n = 175). USDA-certified organic on-farm locations were WP Rawl and Sons (Pelion, SC, USA) and Calhoun Fields Laboratory (Clemson University, SC, USA). Before planting, fields were tilled using a disc harrow and smoothly leveled. The plot size was 1.4×6 m (8.4 m$^2$) with seven rows spaced 20 cm apart, a seeding depth of 5–7 cm, and a seeding rate of 90 seeds m$^{-2}$. USDA-certified organic inoculant (Peaceful Valley Farm Supply, Inc., USA) was added at 3.1 g kg$^{-1}$ seeds. At physiological maturity (110–115 days after planting), the plots were harvested, and 500 g of seeds were hand cleaned, finely ground using a UDY grinder, and stored at −10˚C until protein quality analysis. All protein quality data are reported on a dry mass (15% moisture).

### Protein analysis

Total seed N concentration was measured using N combustion at the Soil Testing Laboratory, Clemson University, SC, and then values were converted to total protein content by multiplying by 6.25. Protein data are reported in our previous publication [5].

## Amino acid (AA) analysis

Reagents, solvents, and high-purity standards for AA analysis were purchased from Sigma Aldrich Co. (St. Louis, MO), Fisher Scientific (Waltham, MA), and VWR International (Radnor, PA). Ultrapure water and deionized water (ddH$_2$O) to a resistance of $\geq$18.2 M$\Omega\times$cm (PURELAB flex 2 system, ELGA LabWater North America, Woodridge, IL) were used. The AA analysis method is reported elsewhere [25] with modifications from the literature [26,27]. Samples (40 mg) of dry pea powder (particle size $\leq$ 0.5 mm) were weighed into glass culture tubes (16$\times$125 mm, PTFE lined cap). Performic acid was synthesized from formic acid and hydrogen peroxide (9:1 ratio). Once chilled in an ice bath, 5 mL of performic acid was added to each tube, which was then gently swirled on a vortex mixer before being capped and refrigerated for 16 h to convert Cys and Met to their derivatives, methionine sulfone and cysteic acid, which are more stable under acid hydrolysis. A 1/8 in. $\times$ tube length PTFE boiling rod was inserted into each tube before evaporating to dryness in a vacuum oil bath (3 gal. resin trap, BACOENG, Suzhou, China) at ~70–80˚C and ~610 mmHg. Once cooled, tubes were removed, and 4.9 mL of 6 M HCl and 0.1 mL of the standard internal mix (25 mM norvaline, 25 mM sarcosine) were added to each tube, which was then capped and gently swirled. Tubes were then placed in a gravity convection oven at 110˚C for 24 h to hydrolyze peptide bonds. Samples were cooled to room temperature, vortex mixed, and filtered through a 0.22 μm polypropylene syringe filter. As before, one mL of the sample was added to a clean culture tube and evaporated to dryness. Samples were rehydrated with 1 mL of HPLC mobile phase A and pipetted into HPLC vials for analysis.

AA analysis was performed via high-performance reverse phase chromatography on an 1100 series Agilent system (Agilent Technologies, Santa Clara, CA, USA) [28,29] with a diode array detector at two wavelengths (338 nm, 10 nm bandwidth, reference 390 nm, 20 nm bandwidth; and 262 nm, 10 nm bandwidth, reference 390 nm, 20 nm bandwidth). An aqueous and an organic solvent were used for mobile phases A and B, respectively. Mobile phase A contained 10 mM sodium phosphate, 10 mM sodium tetraborate decahydrate, and 5 mM sodium azide with a pH adjusted to 8.2 with 12 M HCl. The solution was then filtered through 0.2 μm regenerated cellulose. Mobile phase B consisted of 45% methanol, 45% acetonitrile, and 10% water (v/v/v). A lab reference dry pea sample was included in every digestion batch to monitor batch-to-batch variation; an AA standard mix was run on the high-performance liquid chromatography (HPLC) before analyzing each batch of samples. Calibration standards (9–900 pmol/μL) with internal standards norvaline and sarcosine (500 pmol/μL) were run, and linear calibration models were generated based on peak areas for calculating sample AA concentrations, which were converted into percent of dry pea flour. The total AA concentration was calculated by summing all AA concentrations for each sample.

## In vitro protein digestibility analysis

Protein digestibility was measured using the Megazyme Protein Digestibility Amino Acid Score assay kit with the modified protocol for a 100 mg sample size (Megazyme K-PDKAAS kit, Lancing, MI). Ground samples (100 mg) were weighed into 50 mL plastic falcon tubes, to which 3.8 mL of 0.06 N hydrochloric acid was added, and the mixture vortexed. The tubes were then placed into a tabletop heated air shaker at 37˚C for 30 min at 300 rpm. After shaking, 0.2 mL of pepsin solution was added to the tube, and the mixture vortexed. The tubes were then placed back into the shaker at 37˚C for 60 min at 300 rpm, and after the pepsin incubation, 0.4 mL of TRIS buffer was added. The tubes were then vortexed, and 40 μL of trypsin/chymotrypsin solution was added to the tubes and then placed back in the air shaker for 4 h. After the trypsin/chymotrypsin incubation, the tubes were placed in a 100˚C water bath for 10

min and then vortexed and brought to room temperature on the counter for a minimum of 20 min. After the overnight cold incubation, the tubes were centrifuged for 10 min. Ninety-six well plates were utilized for the colorimetric analysis. The Megazyme Excel calculator was modified to change the approximate sample mass from 0.5 to 0.1 g. In addition to the controls in the assay kit, a lab reference lentil sample was included in every batch to monitor batch-to-batch variation. The protein digestibility corrected amino acid score (PDCAAS) was calculated based on the Megazye Excel calculator, determined by comparing the AA profile of the dry pea against a standard AA profile, with one as the highest possible score.

## Statistical analysis

Replicates, years, and cultivars were used as class variables. Data from both years were combined (after testing for heterogeneity) and analyzed using a general linear model procedure (PROC GLM) mixed model. Fisher's least significant difference (LSD) at $\leq 0.05$ was performed for mean separation. Correlations (Pearson correlation coefficients) among traits were determined. A statistical model was developed to estimate broad-sense heritability ($H^2$) with the class variables and genotype as random effects. The model was fit using restricted maximum likelihood (REML). $H^2$ was estimated as the proportion of variance due to cultivar, and analyses were performed using JMP 14.0.0 and SAS 9.4 [30]. Percent recommended dietary allowance estimates (%RDA) were calculated for the essential AAs [Cys, histidine (His), isoleucine (Iso), leucine (Leu), Lys, Met, phenylalanine (Phe), threonine (Thr), valine (Val)] and total AA concentration. Estimates were based on a 72 kg adult consuming 100 g of dry pea (15% moisture content) per day: 8–12 mg/kg His, 10 mg/kg Iso, 14 mg/kg Leu, 12 mg/kg Lys, 13 mg/kg Met + Cys, 14 mg/kg Phe + Tyr, 10 mg/kg Val, and 0.8 g/kg protein [31].

## Results

### Analysis of variance

Cultivars showed significant variation at *P<0.05* and *P<0.1* for most traits except for His, hydroxyproline (Hpr), Lys, and *in vitro* protein digestibility (Table 1). Location was significant for most cases except for serine (Ser) and total AAs. Similarly, the year effect was significant at *P<0.05* and *P<0.1* for 12 of 17 AAs, total AAs, total protein, and *in vitro* digestibility. Significant interactions of either cultivar × location or cultivar × year varied with the traits. The *in vitro* protein digestibility showed a significant effect only with the location and year; no effect was evident with cultivar × location or cultivar × year (Table 1). Broad-sense heritability estimates were very low to moderate (0.06–0.42), with the highest for arginine (Arg; 0.42) and total protein (0.25). Broad-sense heritability estimates were very low for SAAs (Met and Cys) and Lys (Table 1).

### Protein quality

Organic dry pea cultivars had values of 11.8 to 22.2 g/100 g for total AAs (liberated), 12.6 to 27.6 g/100 g for total protein, 0.18 to 0.64 for PDCAAS value, and 83 to 95% for *in vitro* protein digestibility (Table 2). Dry pea contained a range of individual AAs, including nine essential AAs with a mean of 0.22 g/100 g for SAAs and 0.88 g/100 g for Lys (Table 2). These organic dry pea cultivars provide a significant amount of the recommended daily allowance (%RDA) of several AAs (14–66% His, 79–138% Iso, 76–169% Leu, 57–147% Lys, 15–85% Met + Cys, 76–189% Phe + Tyr, 94–169% Val) as well as protein (22–48%) (Table 2). Pearson's correlation analysis revealed that most correlations were significantly positive except for Hpr vs. His and *in vitro* protein digestibility vs. Lys (Table 3). Total protein showed a significant positive

**Table 1. Analysis of variance and broad-sense heritability estimates of protein quality traits evaluated for dry pea tested in SC, USA.**

| Component | Cultivar | Location | Year | Cultivar × Location | Cultivar × Year | $H^2$ |
|---|---|---|---|---|---|---|
| Alanine | ** | ** | * | ** | ** | 0.11 |
| Arginine | ** | ** | ** | NS | * | 0.42 |
| Asparagine | ** | ** | ** | ** | ** | 0.08 |
| Cystine | * | ** | NS | ** | ** | - |
| Glutamine | ** | ** | NS | ** | ** | 0.24 |
| Glycine | ** | * | ** | ** | ** | 0.19 |
| Histidine | NS | ** | NS | NS | NS | 0.14 |
| Hydroxyproline | NS | ** | ** | NS | NS | - |
| Isoleucine | ** | ** | ** | ** | ** | 0.23 |
| Leucine | ** | ** | ** | ** | ** | 0.18 |
| Lysine | NS | * | ** | NS | NS | 0.17 |
| Methionine | ** | * | NS | NS | NS | 0.12 |
| Phenylalanine | * | ** | NS | ** | * | 0.23 |
| Proline | ** | ** | ** | * | ** | 0.18 |
| Serine | ** | NS | ** | ** | ** | 0.13 |
| Threonine | ** | * | ** | ** | ** | 0.06 |
| Valine | ** | ** | ** | ** | ** | 0.13 |
| Total AA | ** | NS | ** | ** | ** | 0.19 |
| Total Protein | ** | ** | ** | NS | * | 0.25 |
| In-vitro Digestibility | NS | ** | ** | NS | NS | 0.09 |

** significant at $P<0.05$;

* significant at $P<0.1$;

Not significant (NS); $H^2$ broad-sense heritability estimate.

correlation with all AAs except Hpr; Lys and Cys were also not correlated; and Hpr showed non-significant correlations in several cases (Table 3). The first two principal components (PCA) of the principal component analysis (PCA) accounted for 12.46, and 1.83 for the eigen-values. Cluster summary showed components of the total variance: (1) component 1 (62.3%): total AAs and 13 of 17 AAs; (2) component 2 (9.17%): Hpr and His; (3) component 3 (8.07%): *in vitro* protein digestibility; (4) component 4 (5.34%); protein and Arg; and (5) component 5 (2.93%): Cys (Fig 1). Most of the variation was captured by the first component (62.3%), which is highly correlated with the values of most AAs excluding Hpr and His.

## Cultivar responses

Dry pea cultivars showed a normal distribution pattern for Cys, Met, total AAs, and *in vitro* protein digestibility (Fig 2). Out of 175 observations, 6.4% were high in Cys and Met, 8.8% were high in total AAs, and 5.6% were high for *in vitro* protein digestibility (Fig 2). Among the 25 cultivars tested, 10 cultivars showed more than 18 g/100 g of total AAs, with Fiddle being the highest and AAC Carver and AC Earlystar the lowest (Fig 3). For *in vitro* protein digestibility, 17 of 25 cultivars showed a digestibility of 87% or better, with Fiddle having the highest value and AAC Carver the lowest (Fig 3). CDC Saffron, CDC Spectrum, and CDC Striker showed significantly higher concentrations of SAAs than AAC Carver and AC Earlystar (Fig 4). AAC Comfort showed higher Lys concentrations than other cultivars, but the effects were not significant (Fig 4).

**Table 2. Range and mean amino acid concentrations of organic dry pea grown in SC.**

| Composition (g/100 g) | Range | Mean | Genotype Effect | %RDA |
|---|---|---|---|---|
| Alanine | 0.61–1.01 | 0.86 | ** | |
| Arginine | 0.95–2.22 | 1.5 | ** | |
| Asparagine | 1.59–3.07 | 2.36 | ** | |
| Cystine | 0.02–0.10 | 0.05 | * | 15–85[‡] |
| Glutamine | 1.82–3.56 | 2.86 | ** | |
| Glycine | 0.60–1.08 | 0.88 | ** | |
| Histidine | 0.08–0.38 | 0.26 | NS | 14–66 |
| Hydroxyproline | 0.48–2.00 | 1.16 | NS | |
| Isoleucine | 0.57–0.99 | 0.8 | ** | 79–138 |
| Leucine | 0.77–1.70 | 1.33 | ** | 76–169 |
| Lysine | 0.49–1.27 | 0.88 | NS | 57–147 |
| Methionine | 0.12–0.26 | 0.17 | ** | |
| Phenylalanine | 0.38–1.16 | 0.89 | * | 76–189 |
| Proline | 0.42–1.32 | 1.04 | ** | |
| Serine | 0.58–1.09 | 0.89 | ** | |
| Threonine | 0.39–0.74 | 0.59 | ** | |
| Valine | 0.68–1.22 | 0.97 | ** | |
| Total AA (liberated) | 11.8–22.2 | 17.5 | ** | |
| Total Protein[±] | 12.6–27.6 | 20.9 | ** | 22–48 |
| PDCAAS value | 18–64 | 54 | ND | |
| *In vitro* digestibility (%) | 83–95 | 87 | NS | |

** significant at $P<0.05$;

* significant at $P<0.1$;

Not significant (NS); ND: Not detected; PDCAAS: Protein digestibility corrected amino acid score.

[±] Protein values are from [4]. Values are based on the combined statistical analysis of 175 data points for the current study (dry weight basis). Percent recommended dietary allowance estimates were calculated for the essential amino acids cystine (Cys), histidine (His), isoleucine (Iso), leucine (Leu), lysine (Lys), methionine (Met), phenylalanine (Phe), threonine (Tyr), and valine (Val), as well as for total AA concentration. Estimates were for a 72 kg adult consuming 100 g of dry pea (15% moisture content) per day given the following dietary requirements: 8–12 mg/kg His, 10 mg/kg Iso, 14 mg/kg Leu, 12 mg/kg Lys, 13 mg/kg Met + Cys, 14 mg/kg Phe + Tyr, 10 mg/kg Val, and 0.8 g/kg protein [31].

[‡] %RDA was calculated for both Cys+Met.

## Discussion

Our results demonstrate that current dry pea cultivars bred for conventional systems vary in terms of seed AA profile, total AAs, total protein, and *in vitro* protein digestibility when grown under organic cropping systems. Organic dry pea is a rich source of essential AAs, as a 100 g serving of organic dry pea provides 0.02–3.07 g/100 g of nine essential AAs (14–180% of RDA), 11.8–22.2 g of total AAs, and 22–48% of the daily protein requirement, with an *in vitro* protein digestibility of 83–95% (Table 2). In contrast to previous literature that states pulses are generally low in SAAs, our results demonstrate organic dry pea is a good source of SAAs (Met and Cys), with a 100 g serving providing 220 mg of total SAAs (Met+Cys) and 0.88 g of Lys (Table 2; Fig 4) [32,33]. According to our knowledge, this study is the first report on the detailed protein quality of commercial dry pea cultivars grown in an organic system to adapt these cultivars for organic produciton.

The dry pea cultivars tested under organic field conditions in this study had mean protein and total AA (liberated) concentrations of 20.9 g/100 g and 17.5 g/100 g, respectively (Table 2).

**Table 3. Pearson's correlation analysis of nutritional traits among dry pea cultivars grown in the organic system.**

| | Cys | Asp | Glu | Ser | His | Gly | Thr | Met | Arg | Ala | Val | Phe | Iso | Leu | Lys | Hpr | Pro | AA | Pr | Dig |
|---|---|---|---|---|---|---|---|---|---|---|---|---|---|---|---|---|---|---|---|---|
| Cys | - | | | | | | | | | | | | | | | | | | | |
| Asp | ** | - | | | | | | | | | | | | | | | | | | |
| Glu | ** | ** | - | | | | | | | | | | | | | | | | | |
| Ser | ** | ** | ** | - | | | | | | | | | | | | | | | | |
| His | ** | ** | ** | ** | - | | | | | | | | | | | | | | | |
| Gly | ** | ** | ** | ** | ** | - | | | | | | | | | | | | | | |
| Thr | ** | ** | ** | ** | ** | ** | - | | | | | | | | | | | | | |
| Met | ** | ** | ** | ** | ** | ** | ** | - | | | | | | | | | | | | |
| Arg | ** | ** | ** | ** | ** | ** | ** | ** | - | | | | | | | | | | | |
| Ala | ** | ** | ** | ** | ** | ** | ** | ** | ** | - | | | | | | | | | | |
| Val | ** | ** | ** | ** | ** | ** | ** | ** | ** | ** | - | | | | | | | | | |
| Phe | ** | ** | ** | ** | ** | ** | ** | ** | ** | ** | ** | - | | | | | | | | |
| Iso | ** | ** | ** | ** | ** | ** | ** | ** | ** | ** | ** | ** | - | | | | | | | |
| Leu | ** | ** | ** | ** | ** | ** | ** | ** | ** | ** | ** | ** | ** | - | | | | | | |
| Lys | NS | ** | ** | ** | ** | ** | ** | ** | ** | ** | ** | ** | ** | ** | - | | | | | |
| Hpr | NS | ** | * | ** | -** | ** | ** | ** | NS | NS | ** | NS | NS | ** | NS | - | | | | |
| Pro | ** | ** | ** | ** | ** | ** | ** | ** | ** | ** | ** | ** | ** | ** | ** | ** | - | | | |
| Total AA | ** | ** | ** | ** | ** | ** | ** | ** | ** | ** | ** | ** | ** | ** | ** | ** | ** | - | | |
| Total Protein± | ** | ** | ** | ** | ** | ** | ** | ** | ** | ** | ** | ** | ** | ** | ** | NS | ** | ** | - | |
| Digestibility | ** | ** | ** | ** | * | ** | ** | ** | ** | ** | ** | ** | ** | ** | -** | ** | ** | ** | ** | - |

** significant at *P<0.05*;

Not significant (NS);

± Protein values are from [4].

Several dry pea cultivars had high total AAs (>18 g/100 g) and >87% *in vitro* protein digestibility (Fig 3), demonstrating they are suitable for organic plant-based protein production. Among the cultivars tested, Fiddle had the highest total AA concentrations (19.6 g/100 g), and AAC Carver (15.5 g/100 g) and AC Earlystar (16.1 g/100 g) the lowest. Our previous study on the agronomic adaptability of dry pea [5] indicated AAC Carver, Jetset, and Mystique as the highest yielding cultivars (>2000 kg/ha) and most suitable for organic production without a yield penalty compared to conventional growing systems. However, the current study indicates

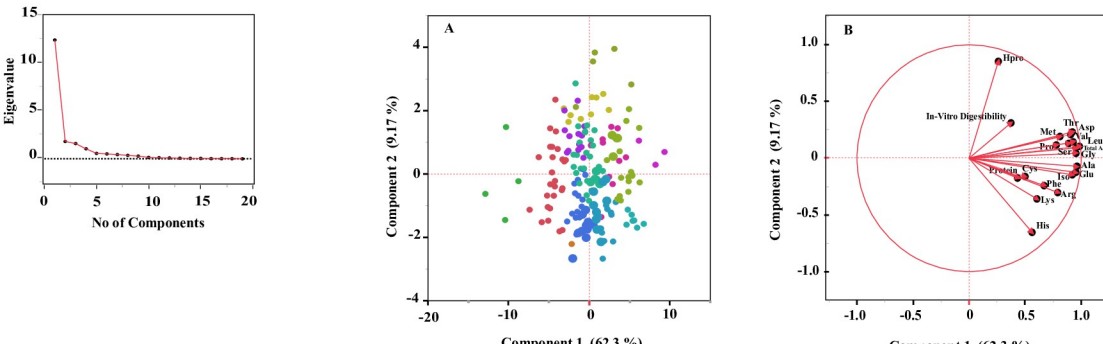

**Fig 1. Principal components of individual amino acids (g/100 g), total amino acids (g/100 g), protein (g/100 g), and *in vitro* digestibility of organic dry pea: (A) scatter plots and (B) biplots of components 1 and 2.**

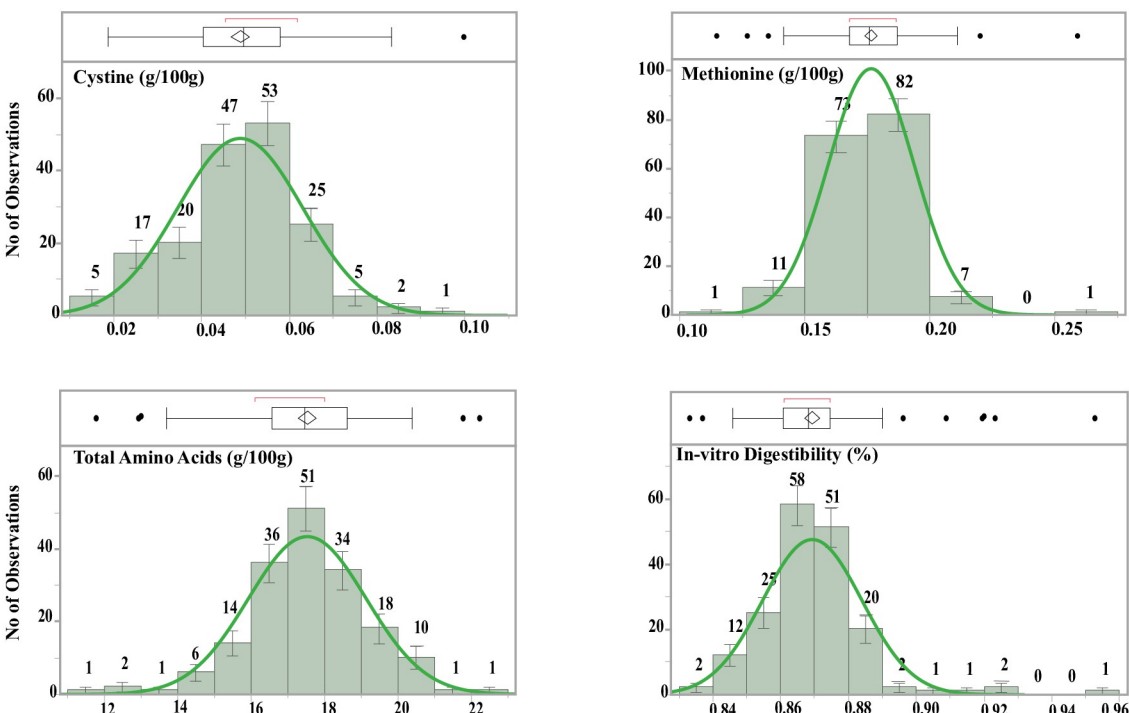

**Fig 2. Dry pea cultivar distribution for cystine, methionine, and total amino acid (liberated) concentration as well as *in vitro* digestibility.**

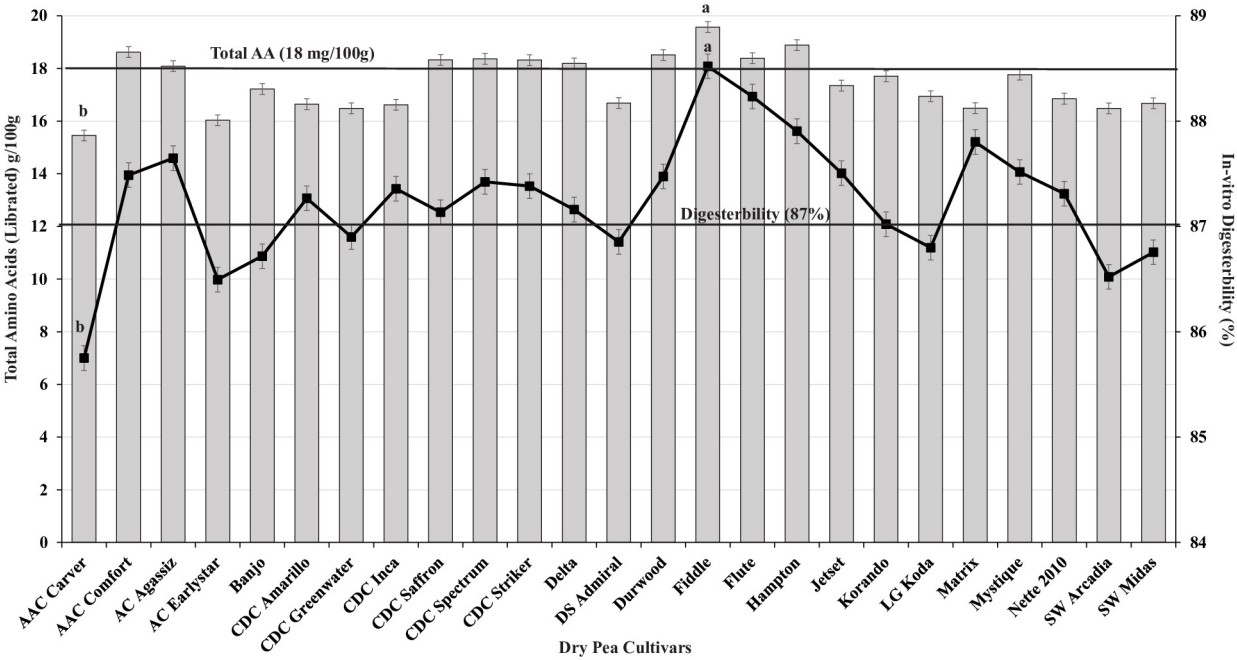

**Fig 3. Total amino acids (liberated, g/100 g) and *in vitro* protein digestibility (%) of dry pea cultivars grown in the organic system.** The bars indicate the total amino acids (liberated) g/100g and the line indicates in-vitro digesterbility (%). The bars/lines with different letters indicate significant difference at *P<0.05*.

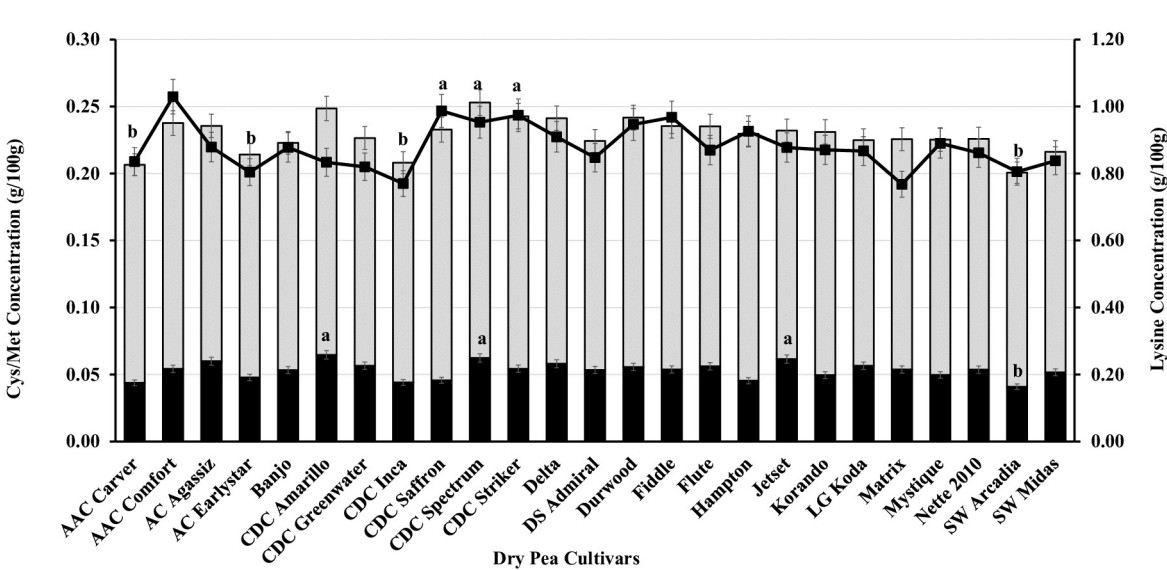

**Fig 4. Organic dry pea cultivar genetic variation for seed cystine, methionine, and lysine concentrations.** The bars/lines with different letters indicate a significant difference at $P<0.05$.

these three cultivars have low total AAs and *in vitro* protein digestibility (Fig 3). A negative correlation between protein quality and crop adaptability suggests further testing is needed with diverse dry pea germplasm to develop biofortified organic cultivars with better grain yield, agronomic adaptability, and protein quality for organic systems [5,9,34]. Earlier literature [32,35] indicates the AA composition of dry pea varies with cultivar and growing environment, similar to the current study's results. Further, one of these earlier studies shows dry pea has high concentrations of Arg, Leu, Lys, aspartic acid, and glutamic acid and low concentrations of His, Met, Thr, and Cys [35]. Another study compared several plant-based protein isolates for essential and non-essential AAs and found dry pea protein isolates contained only 5.9% Lys and low concentrations of Met [36]. In contrast, our study results show most modern cultivars have higher Cys, Met, and total AA concentrations and good *in vitro* protein digestibility (Fig 3). The best options to use for better protein quality are CDC Spectrum for Met and Cys, CDC Inca and CDC Striker for Met, and CDC Amarillo and Jetset for Cys (Fig 4). These cultivars have AA values within the range of the AAs reported in the literature for conventional cropping systems [33,36]. Incorporating these cultivars into dry pea breeding programs would benefit the development of better protein quality cultivars; however, more field testing is required to understand the genetic, environmental, and management interactions. Organic agriculture management varies with respect to on-farm practices for weeds, diseases, pests, and fertilizer; therefore, breeding dry pea cultivars best suited for organic management with increased nutritional quality is challenging [37].

AAs are critical for all forms of life. Humans cannot synthesize all 20 AAs needed for protein synthesis for good health. Nine essential AAs must be obtained from the diet: Lys, Met, and Thr of the aspartate (Asp) family pathway; phenylalanine (Phe) and tryptophan (Trp) of the aromatic AAs; Val, Ile, and Leu of the branched-chain Aas (BCAAs); and His [38]. Lys, Met, Thr, and Trp levels limit the nutritional quality of plant-based foods because levels of these four AAs in plants are very low compared with those required for optimal human

nutrition [9,38]. PCA analysis in the current study revealed seven essential AAs (Val, Iso, Thr, Leu, Met, Lys, and Phe) of organic dry pea in component 1, and one essential AA (His) in component 2 (Fig 1). These essential AAs are also positively correlated with total AA, protein, and *in vitro* digestibility (Table 3), indicating biosynthesis of these AAs could be upregulated using available genomic and biotechnology tools for early prediction of protein quality traits in breeding programs [9,38,39]. Plant-soluble Met and Lys levels might represent limiting factors for synthesizing Met- or Lys-rich proteins [37]. Expressing genes that increase Lys and Met biosynthesis in combination with genes encoding proteins rich in Lys and Met codons appears to increase the levels of Lys in transgenic corn [39]. However, these transgenic approaches are not approved in USDA-certified organic agriculture systems. Conventional breeding approaches for selecting genetic material with higher levels of AAs and protein quality using association mapping and genomic prediction tools are the only recommended methods for organic pulse breeding.

Dietary protein quality has two components: AA composition and availability. Availability is "the proportion of the dietary amino acids that are digested and absorbed in a form suitable for body protein synthesis" [40]. PDCAAS is the most common method used to determine protein availability [41]. We determined *in vitro* protein digestibility using an enzyme assay and then calculated PDCAAS based on the AA scores. This method is inexpensive and high-throughput and can be used to screen a larger number of seed samples for breeding programs than available *in vivo* methods [42]. The PDCAAS values for organic dry pea cultivars tested in this study ranged from 0.18 to 0.64 with 83–95% *in vitro* protein digestibility. Most organic dry pea cultivars have high protein digestibility (>87%), and these values are similar to those from the literature [43]. Plant-based proteins are an inexpensive, healthy choice for many people and a vital source of daily essential AAs. These proteins have several limitations in terms of human nutrition: they often lack one or more essential AAs, they are often not fully digestible, and toxins and pesticides are concentrated during protein extraction and drying procedures [44]. Therefore, pursuing nutritional breeding or biofortification of dry pea using an organic system approach is vital to overcome these nutritional and production issues for pulse growers and consumers. Organic nutritional breeding of pulses is challenging and demands better phenotyping and genetic resources for cultivar development. With the increasing availability of genomic resources, expanding organic pulse breeding targets to produce better-quality proteins with higher digestibility will be possible.

## Supporting information

**S1 Data.**
(XLSX)

## Acknowledgments

We thank the Crop Improvement Team (Clemson University), David Robb (Organic Farm, Clemson University), and Charles Wingard and Ashely Rawls from WP Rawl & Sons, Inc. for field operations; Martin Hochhalter (Meridian Seeds), Tyler Kres (Pulse USA), and the Washington State Crop Improvement Association for providing dry pea seeds; and Elizabeth Bean, Jacob Johnson, Lindsey Moroney and Richard Baker for the technical assistant for the pulse breeding program.

## Author Contributions

**Conceptualization:** Dil Thavarajah, Emerson Shipe, Shiv Kumar.

**Data curation:** Dil Thavarajah, Tristan Lawrence, Lucas Boatwright, Nathan Windsor, Nathan Johnson, Joshua Kay, Emerson Shipe, Shiv Kumar, Pushparajah Thavarajah.

**Formal analysis:** Dil Thavarajah, Nathan Johnson, Shiv Kumar.

**Funding acquisition:** Dil Thavarajah, Lucas Boatwright, Emerson Shipe, Shiv Kumar.

**Investigation:** Dil Thavarajah, Shiv Kumar.

**Methodology:** Dil Thavarajah, Tristan Lawrence, Nathan Windsor, Nathan Johnson, Joshua Kay, Pushparajah Thavarajah.

**Project administration:** Dil Thavarajah.

**Resources:** Dil Thavarajah.

**Software:** Dil Thavarajah, Nathan Windsor, Nathan Johnson, Joshua Kay.

**Supervision:** Dil Thavarajah.

**Validation:** Dil Thavarajah, Nathan Johnson, Shiv Kumar, Pushparajah Thavarajah.

**Visualization:** Dil Thavarajah.

**Writing – original draft:** Dil Thavarajah.

**Writing – review & editing:** Dil Thavarajah, Tristan Lawrence, Lucas Boatwright, Emerson Shipe, Shiv Kumar, Pushparajah Thavarajah.

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
