## [Decision Letter · Decision Letter 0]

8 Nov 2022

PONE-D-22-26254Organic dry pea (Pisum sativum L.): a sustainable alternative pulse-based protein for human healthPLOS ONE

Dear Dr. Thavarajah,

Thank you for submitting your manuscript to PLOS ONE. After careful consideration, we feel that it has merit but does not fully meet PLOS ONE’s publication criteria as it currently stands. Therefore, we invite you to submit a revised version of the manuscript that addresses the points raised during the review process.

We look forward to receiving your revised manuscript.

Kind regards,

Aditya Pratap

Academic Editor

PLOS ONE

Journal Requirements:

Additional Editor Comments:

I am now in receipt of review reports for your manuscript. As you would notice, both find the MS to be of interest while one of them has been especially critical on some of the aspects which need to be thoroughly addressed while submitting your revised version. Beside this, I also have my own observations which require attention:

1. The topic has been introduced mostly in light of the US markets and cultivation practices while there are several other regions where dry peas are an important crop. Accordingly, the same along with supporting data needs to be mentioned in the 'Introduction' section.

2. The source of consumables may be mentioned within the text as and when required and not as a separate paragraph, that too the first para of the 'M&M' section.

3. While a reference has been cited (lines 129-130) about agronomic practices, etc. it is advised to add a couple of sentences on the same to give the authors a continuity in case they cannot immediately consult the previous paper.

4. Table 1 in not required.

Reviewers' comments:

Reviewer's Responses to Questions

**Comments to the Author**

1. Is the manuscript technically sound, and do the data support the conclusions?

Reviewer #1: Partly

Reviewer #2: Yes

2. Has the statistical analysis been performed appropriately and rigorously? 

Reviewer #1: No

Reviewer #2: Yes

3. Have the authors made all data underlying the findings in their manuscript fully available?

Reviewer #1: Yes

Reviewer #2: Yes

4. Is the manuscript presented in an intelligible fashion and written in standard English?

Reviewer #1: Yes

Reviewer #2: Yes

5. Review Comments to the Author

Reviewer #1: The focus of this paper is on organic pea. The authors should speculate on whether their results would have differed had the research been conducted under conventional agriculture instead of organic. As organic pea production is a very small industry in the world, the authors should comment on whether their results could be more broadly applicable to conventional agriculture. What are the novel findings in this paper other than organic?

My specific comments:

The abstract gives the impression that the field studies were conducted over 4 station-years (2 years X 2 locations), however, in reality it was only 3 station-years. This should be made clear. Three station-years is quite minimal for field based agricultural research and this limitation should be mentioned in the paper. With only three station years conducted, I recommend that the data are reanalysed in that way instead of trying to separate the effects of year and location, since the number of years and locations was minimal.

Line 38. PDCAAS is usually expressed as a number between 0 and 1.

63 novel,

66 indicate the number of acres and tonnes of organic pea production in the USA and by region in USA

68 Impossible burger uses soybean protein, not pea protein

93 is not the only indicator ...

107 likely not 'most breeding programs'

109-111 provide a reference for this statement

124 only 3 station-years, so it is a limited dataset

142 if the AA analysis is already published, what is novel in this paper? Or do you mean that the METHOD of AA analysis is already published?

142 why was tyrptophan omitted from this research?

221 please include %RDA for all AAs, especially met.

254 the cultivars are not 'organic', the study was conducted under organic conditions

270 isolates are different from whole seeds studied here, so it is not a fair comparison

311 'toxins and pesticides' - really?! please provide a reference for this statement.

Reviewer #2: This manuscript hasn't got enough substance/findings to be published in a reputed journal, however findings on variability in AAs and digestibility among dry pea varieties are new and informative, that's why I am recommending it to be published.

6. PLOS authors have the option to publish the peer review history of their article (what does this mean?). If published, this will include your full peer review and any attached files.

Reviewer #1: No

Reviewer #2: No

---

## [Author Response · Author response to Decision Letter 0]

17 Dec 2022

Response to Reviewers: 

Journal Requirements:

Response: Action has been taken. 

Response: Action has been taken. 

Response: 

Founding Agencies: 

1. National Institute of Food and Agriculture, Organic Agriculture Research and Extension Initiative (OREI) (award no. 2018-51300-28431/proposal no. 2018-02799) and the United States Department of Agriculture,

2. National Institute of Food and Agriculture, Organic Agriculture Research and Extension Initiative (OREI) (award no. 2021-51300-34805/proposal no. 2021-02927) (DT, LB) 

3. USDA National Institute of Food and Agriculture, [Hatch] project [1022664] (DT); 

4. Good Food Institute (DT) 

5. FoodShot Global (DT) 

b) State what role the funders took in the study. If the funders had no role in your study, please state: 

Response: Funders have no role in this study: The funders had no role in study design, data collection, and analysis, decision to publish, or preparation of the manuscript.

Response: No salary was received from funders.

Response: No action is needed. 

Response: Action has been taken – all the information was added to the cover letter. 

Response: All data used for this manuscript is available as a supplementary file attached to the manuscript. 

Action taken. 

Action taken. 

Additional Editor Comments:

I am now in receipt of review reports for your manuscript. As you would notice, both find the MS to be of interest while one of them has been especially critical on some of the aspects which need to be thoroughly addressed while submitting your revised version. Beside this, I also have my own observations which require attention:

1. The topic has been introduced mostly in light of the US markets and cultivation practices while there are several other regions where dry peas are an important crop. Accordingly, the same along with supporting data needs to be mentioned in the 'Introduction' section.

Response: Please see the new paragraph line 66-73. 

2. The source of consumables may be mentioned within the text as and when required and not as a separate paragraph, that too the first para of the 'M&M' section.

Response: Removed the paragraph and added it to the text. 

3. While a reference has been cited (lines 129-130) about agronomic practices, etc. it is advised to add a couple of sentences on the same to give the authors a continuity in case they cannot immediately consult the previous paper.

Response: Please see the new lines 133-134. 

4. Table 1 in not required.

Response: Table 1 has been removed. 

Reviewers' comments:

Reviewer's Responses to Questions

Comments to the Author

1. Is the manuscript technically sound, and do the data support the conclusions?

Reviewer #1: Partly

Reviewer #2: Yes

2. Has the statistical analysis been performed appropriately and rigorously? 

Reviewer #1: No

Reviewer #2: Yes

3. Have the authors made all data underlying the findings in their manuscript fully available?

Reviewer #1: Yes

Reviewer #2: Yes

4. Is the manuscript presented in an intelligible fashion and written in standard English?

Reviewer #1: Yes

Reviewer #2: Yes

5. Review Comments to the Author

Reviewer #1: The focus of this paper is on organic pea. The authors should speculate on whether their results would have differed had the research been conducted under conventional agriculture instead of organic. As organic pea production is a very small industry in the world, the authors should comment on whether their results could be more broadly applicable to conventional agriculture. What are the novel findings in this paper other than organic?

Response: Thank you for the comments. A sentence will be added to the discussion. 

My specific comments:

The abstract gives the impression that the field studies were conducted over 4 station-years (2 years X 2 locations), however, in reality it was only 3 station-years. This should be made clear. Three station-years is quite minimal for field based agricultural research and this limitation should be mentioned in the paper. With only three station years conducted, I recommend that the data are reanalyzed in that way instead of trying to separate the effects of year and location since the number of years and locations was minimal.

Response: thank you for the response. Clarity was included in the abstract and the method section. We did not separate the year and location effect, and data were analyzed as combined (lines 190-192). 

Line 38. PDCAAS is usually expressed as a number between 0 and 1.

Response: Thank you, it was corrected. Sorry for the typo. New lines 187-189.

63 novel,

Response: corrected. 

66 indicate the number of acres and tonnes of organic pea production in the USA and by region in USA

Response: New lines 66-71. Organic dry pea acreage is not available yet. 

68 Impossible burger uses soybean protein, not pea protein

Response: Thank you for the correction, yes it was removed. 

93 is not the only indicator ...

Response: corrected. 

107 likely not 'most breeding programs'

Response: corrected

109-111 provide a reference for this statement

Response: Reference is added. 

124 only 3 station-years, so it is a limited dataset.

Response: Thank you for the comments. I agree that it is better to have more locations and years but with limited funding availability for our research most pre-breeding studies are conducted for two years and use selected parents for speed breeding. 

142 if the AA analysis is already published, what is novel in this paper? Or do you mean that the METHOD of AA analysis is already published?

Response: We published the AA analysis method not the data. Please see new line 150.

142 why was tyrptophan omitted from this research?

Response: Tryptophan is breakdown during acid hydrolysis and concentrations will be below the detection limits. Therefore, we are not reporting tryptophan. 

221 please include %RDA for all AAs, especially met.

Response: %RDA can be calculated only for the estimates that are available for His, Iso, Lue, Lys, Met+Cys, Phe+Tyr and Val only from reference 31. It is not able to calculate individually for the Met and Cys. Methionine values are the same as cystine values as it gives a sum of Met and Cys. Please refer to reference 31 for the RDA values that are only available for listed essential AA. Please see the footnote in the Table 2. 

254 the cultivars are not 'organic', the study was conducted under organic conditions

Response: Corrected – thank you. 

270 isolates are different from whole seeds studied here, so it is not a fair comparison

Response: Agree, but isolates were extracted from the whole seed. It is good information to include in the discussion. 

311 'toxins and pesticides' - really?! please provide a reference for this statement.

Response: Yes, that is correct, as glyphosate and other pesticides are concentrated during the protein isolation process. My lab has filed a patent to develop organic proteins without harmful pesticides and chemicals. The patent will be available to the public next year. We go for organic pulse breeding to avoid pesticides and other chemicals in our pea and lentil proteins. I also include a reference. 

Reviewer #2: This manuscript hasn't got enough substance/findings to be published in a reputed journal, however findings on variability in AAs and digestibility among dry pea varieties are new and informative, that's why I am recommending it to be published.

Response: All comments attached to the PDF was corrected. 

6. PLOS authors have the option to publish the peer review history of their article (what does this mean?). If published, this will include your full peer review and any attached files.

Do you want your identity to be public for this peer review? For information about this choice, including consent withdrawal, please see our Privacy Policy.

Reviewer #1: No

Reviewer #2: No

---

## [Decision Letter · Decision Letter 1]

29 Mar 2023

Organic dry pea (Pisum sativum L.): a sustainable alternative pulse-based protein for human health

PONE-D-22-26254R1

Dear Dr. Thavarajah,

We’re pleased to inform you that your manuscript has been judged scientifically suitable for publication and will be formally accepted for publication once it meets all outstanding technical requirements.

Kind regards,

Aamir Raina, Ph.D

Academic Editor

PLOS ONE

Additional Editor Comments (optional):

Reviewers' comments:

Reviewer's Responses to Questions

**Comments to the Author**

1. If the authors have adequately addressed your comments raised in a previous round of review and you feel that this manuscript is now acceptable for publication, you may indicate that here to bypass the “Comments to the Author” section, enter your conflict of interest statement in the “Confidential to Editor” section, and submit your "Accept" recommendation.

Reviewer #1: All comments have been addressed

Reviewer #2: All comments have been addressed

2. Is the manuscript technically sound, and do the data support the conclusions?

Reviewer #1: Yes

Reviewer #2: Yes

3. Has the statistical analysis been performed appropriately and rigorously? 

Reviewer #1: Yes

Reviewer #2: Yes

4. Have the authors made all data underlying the findings in their manuscript fully available?

Reviewer #1: Yes

Reviewer #2: Yes

5. Is the manuscript presented in an intelligible fashion and written in standard English?

Reviewer #1: Yes

Reviewer #2: Yes

6. Review Comments to the Author

Reviewer #1: Thank you for addressing my comments. I support publication of this paper, although the overall data set used (3 station-years) is quite minimal for papers of this nature.

Reviewer #2: This study determines the genetic variation in amino acids profile, protein and in vitro digestibility of dry pea cultivars under organic farming and provides valuable information on nutritional quality. This needs to be published, however, it would have been ideal if nutritional quality from organic production system was compared with the non-organic production systems for the completeness.

7. PLOS authors have the option to publish the peer review history of their article (what does this mean?). If published, this will include your full peer review and any attached files.

Reviewer #1: No

Reviewer #2: No

---

## [Editor Report · Acceptance letter]

3 Apr 2023

PONE-D-22-26254R1 

Organic dry pea (*Pisum sativum* L.): a sustainable alternative pulse-based protein for human health 

Dear Dr. Thavarajah:

I'm pleased to inform you that your manuscript has been deemed suitable for publication in PLOS ONE. Congratulations! Your manuscript is now with our production department. 

Kind regards, 

on behalf of

Dr. Aamir Raina 

Academic Editor

PLOS ONE